# Exchange Speed of Four-Component Nanorotors Correlates with Hammett Substituent Constants [†]

**Yi-Fan Li, Amit Ghosh, Pronay Kumar Biswas, Suchismita Saha and Michael Schmittel \***

Center of Micro- and Nanochemistry and Engineering, Organische Chemie 1, Universität Siegen,
Adolf-Reichwein-Str. 2, D-57068 Siegen, Germany; yifanli920320@gmail.com (Y.-F.L.);
amit.ghosh@uni-siegen.de (A.G.); biswas@chemie-bio.uni-siegen.de (P.K.B.);
saha@chemie-bio.uni-siegen.de (S.S.)
**\*** Correspondence: schmittel@chemie.uni-siegen.de
[†] Dedicated to Prof. Josef Michl on the occasion of his 80th birthday.

**Abstract:** Three distinct four-component supramolecular nanorotors were prepared, using, for the first time, bipyridine instead of phenanthroline stations in the stator. Following our established self-sorting protocol to multicomponent nanodevices, the nanorotors were self-assembled by mixing the stator, rotators with various pyridine head groups, copper(I) ions and 1,4-diazabicyclo[2.2.2]octane (DABCO). Whereas the exchange of a phenanthroline vs. a bipyridine station did not entail significant changes in the rotational exchange frequency, the *para*-substituents at the pyridine head group of the rotator had drastic consequences on the speed: 4-OMe ($k_{298}$ = 35 kHz), 4-H ($k_{298}$ = 77 kHz) and 4-NO$_2$ ($k_{298}$ = 843 kHz). The exchange frequency (log $k$) showed an excellent linear correlation with both the Hammett substituent constants and log $K$ of the copper(I)–ligand interaction, proving that rotator–copper(I) bond cleavage is the key determining factor in the rate-determining step.

**Keywords:** self-sorting; multicomponent rotor; exchange frequency; Hammett correlation





## 1. Introduction

Over the years, researchers have been probing numerous techniques to mimic the function of biological machines, such as ATP synthase [1,2], bacterial flagella [3], histidine kinase [4], etc. During the iterative and systematic optimization of manmade motors [5–7], increasingly better design strategies have been identified to implement intricate movements and various functions derived therefrom [8–11]. Thus, significant progress has been achieved, for instance, in the field of molecular rotors [12–17], gears [18–21], pumps [22–24], walkers [25–27] and caterpillars [28].

To mimic nature's strategy even closer, one has to realize that life preferentially uses multicomponent assembly for building biological machines. Such approach requires a careful balance of weak interactions that allow for sufficient spatiotemporal binding between components during motion. It is unsurprising that, in the arena of artificial multicomponent devices [29–31], examples with sophisticated dynamic motion are still scarce [32]. Because in multicomponent devices the exchange of a single component may lead to drastically different properties, fundamental insights are needed in how structural and electronic variations will impact on the kinetics of motion.

Herein, we demonstrate that speed changes in four-component nanorotors by exclusively varying the rotator head group are linearly correlated with Hammett substituent constants. Whereas Hammett correlations are abundant for describing kinetic reactivity and thermodynamic properties of organic compounds [33], analogous correlations with supramolecular devices remain largely unexplored [34–36], possibly because many of the design strategies are not robust enough to tolerate larger electronic and steric changes. Although at first glance these results appear marginal, they furnish a tool to precisely pre-

dict and tune the frequency of such four-component rotors that, as we know, are catalytic machinery, where the machine speed determines the rate of catalysis [37].

The family of four-component nanorotors [38] has demonstrated, in our hands, great potential in various fields of application, ranging from catalysis [33] to molecular logic [39]. Till now, several strategies to change the rotational frequency have been explored, including adding external brake stones [40,41], changing the flexibility of the rotator arm [42], changing the number of binding sites [33] and adding nucleophilic additives [43]. All of these studies corroborated our initial hypothesis that the rate-determining step depended on the ligand–metal dissociation. Therefore, the kinetic behavior (rotational frequency of the nanorotor) should correlate with thermodynamic data (binding constant of the metal complex). The present work now sought to establish a quantitative relationship by using the well-known Hammett equation as a link between thermodynamic and kinetics. Undoubtedly, the Hammett equation [44] is known as the most important member of a large amount of linear free energy relationships (LFERs) [45].

The four-component nanorotors of this study (Figure 1) were assembled by following our established self-sorting protocol, by combining rotators **1–3** (with various pyridine head groups), stators **4** and **5**, copper(I) ions and 1,4-diazabicyclo[2.2.2] octane (DABCO). In this process, the zinc porphyrin units from the stator and rotator are linked by DABCO, in a hetero-sandwich complex, whereas the copper(I)-filled phenanthroline sites of the stator are additionally connected to the pyridine nitrogen of the rotator (see, for instance, $[Cu_2(\mathbf{1})(\mathbf{4})(DABCO)]^{2+}$) (Figure 1b). The rotators were designed in a way so as to enable electronic influence of the *para*-substituent onto the pyridine head group that is bound to the copper(I) ion sitting in the diimine station of the stator. Since, in the rate-determining step, the $N_{py}\rightarrow[Cu(diimine)]^+$ linkage has to be cleaved, the donor and acceptor qualities of the *para*-substituent were expected to impact on the exchange frequency.

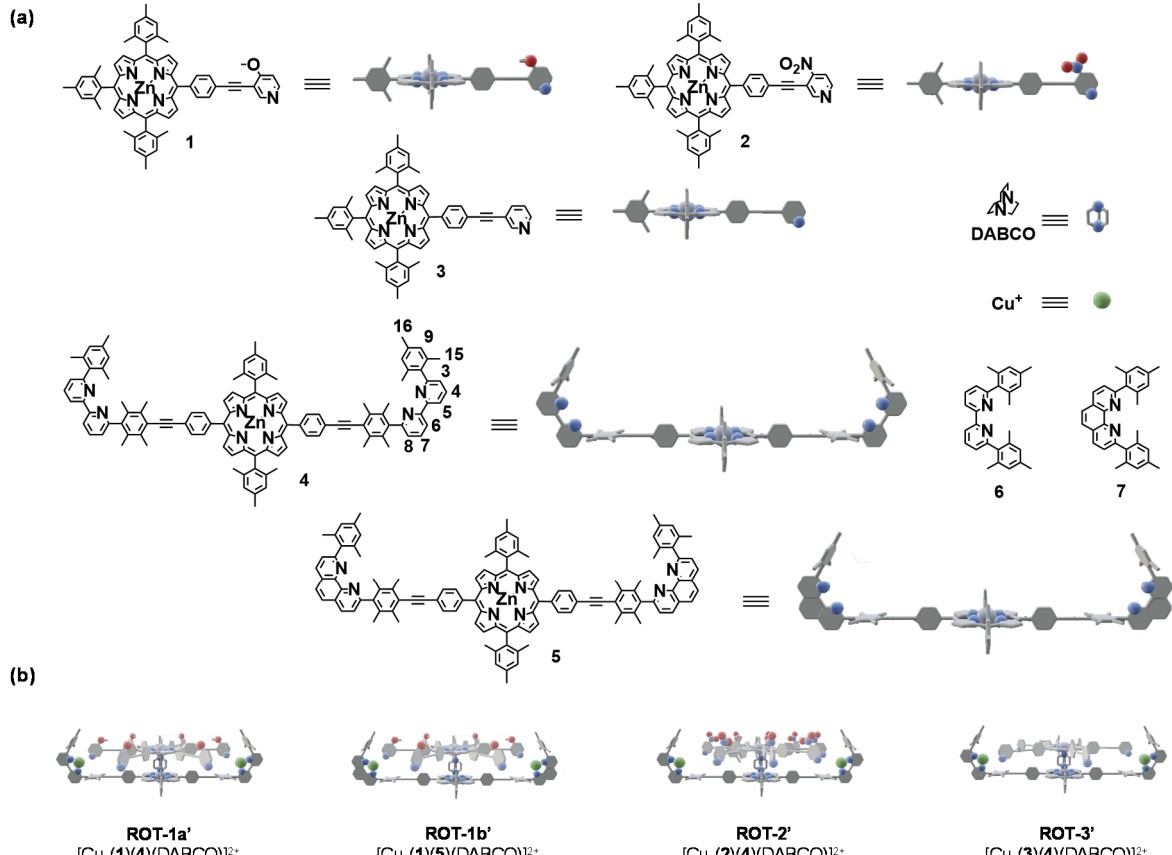

**Figure 1.** (**a**) All four components of the self-assembled nanorotors and their cartoon representation. (**b**) Cartoon representations of the four-component nanorotors.

Both bipyridine and phenanthroline have been amply used as bidentate ligands in supramolecular and coordination chemistry. Compared with phenanthroline, the bipyridine shows less binding strength with metal ions, but higher flexibility for complex reorganization [46]. At the onset, we wanted to evaluate the importance of the flexibility of the bipyridine binding site onto the exchange kinetics of the rotor.

## 2. Materials and Methods

### 2.1. Synthesis

Four nanorotors were synthesized by combining one of the three rotator ligands **1–3** with one of the stators **4** and **5**: **ROT-1a'** = $[Cu_2(1)(4)(DABCO)]^{2+}$, **ROT-1b'** = $[Cu_2(1)(5)(DABCO)]^{2+}$, **ROT-2'** = $[Cu_2(2)(4)(DABCO)]^{2+}$ and **ROT-3'** = $[Cu_2(3)(4)(DABCO)]^{2+}$. For better characterization, equally, the static precursor assemblies containing only one copper(I) ion, here denoted as prerotors, were prepared: **ROT-1a** = $[Cu(1)(4)(DABCO)]^+$, **ROT-1b** = $[Cu(1)(5)(DABCO)]^+$, **ROT-2** = $[Cu(2)(4)(DABCO)]^+$ and **ROT-3** = $[Cu(3)(4)(DABCO)]^+$. All compounds and complexes were characterized by $^1$H-NMR, $^{13}$C NMR, mass spectroscopy and elemental analysis (see Supplementary Materials).

### 2.2. Determination of Binding Constants

UV–Vis titrations were analyzed by fitting the recorded spectra at 0.5 nm intervals, using the SPECFIT® global analysis system by Spectrum Software Associates (Marlborough, MA, USA) [47]. The SPECFIT® program analyzes equilibrium datasets with the help of singular value decomposition and linear regression modeling by the Levenberg–Marquardt method, to determine cumulative binding constants.

### 2.3. NMR Simulation

A conventional dynamic NMR spectroscopic method [48] based on a model involving a two-spin system undergoing mutual exchange was applied to simulate the spectra and determine the exchange frequency. The NMR signal used for the simulation is indicated in the corresponding spectra by an asterisk (*). The exchange frequency that is identical with the rotational frequency was obtained from an analysis of the exchange-broadened NMR signal of proton 16-H of the stator. Activation enthalpy ($\Delta H^\ddagger$) and activation entropy ($\Delta S^\ddagger$) were determined from transition state theory. The temperature-dependent rate constants of the rotors were fitted to the Eyring equation [49]:

$$k = (k_B T/h)e^{-\Delta G\ddagger/RT}, \tag{1}$$

$$\ln(k/T) = -\Delta H^\ddagger/RT + \ln(k_B/h) + \Delta S^\ddagger/R \tag{2}$$

## 3. Results

### 3.1. Design

Previous work about Hammett correlations has mainly focused on mono-substituted aromatic systems, e.g., in equilibria and in fundamental reactions, like hydrolysis of esters, etc. [33]. However, in our rotator's design, the pyridine head groups are disubstituted due to connections to the arm and the probing substituent. It is advisable to attach the rotator's arm in the *meta* position of the pyridine, to minimize any steric hindrance in the HETPYP (heteroleptic pyridine and phenanthroline) [50] complexation to the copper(I) phenanthroline site.

To check whether the Hammett equation applies to the Lewis acid–base interaction of pyridine ($N_{py}$) → $[Cu(bipyAr_2)]^+$, the copper(I)-loaded 6,6'-dimesityl-2,2'-bipyridine $[Cu(bipyAr_2)]^+$, here **C1** = $[Cu(6)]^+$, and its complexes with substituted pyridines $N_{py}$ → $[Cu(bipyAr_2)]$ were studied. As in other HETPYP complexes [50], the steric shielding about the $bipyAr_2$ effectively prevents formation of homoleptic copper bipyridine complexes $[Cu(bipyAr_2)_2]^+$. To quantitatively prepare complex **C1**, $[Cu(CH_3CN)_4](PF_6)$ and **6** (1:1) were simply mixed in $CH_2Cl_2$. The 3-bromo-4-X-pyridine ligands **8–10** were titrated with a standard solution of **C1** = $[Cu(6)]^+$ in $CH_2Cl_2$ and the binding isotherms were

determined by UV–Vis spectroscopy (Figures S58–S60). For comparison, the copper(I)-loaded phenanthroline **C5** = [Cu(**7**)]$^+$ was titrated with ligand **8**, for determining the binding constant in **C6** (Figure S61). The Gibbs free energy, $\Delta G_{298}$, of **C2**, **C3**, **C4** and **C6** was calculated based on the binding constants (Table 1). Both log $K$ and $\Delta G_{298}$ show a linear correlation with $\sigma_p$ (Figure 2).

**Table 1.** Substituent constants, binding constants and Gibbs free energy of **C2–C4** and **C6**, as determined from UV–Vis titrations.

| Motifs | 4-X [a] | $\sigma_p$ | log $K$ | $\Delta G_{298}$ (kJ mol$^{-1}$) |
|---|---|---|---|---|
| **C2** = [Cu(**6**)(**8**)]$^+$ | OMe | −0.268 | 4.59 | −26.2 |
| **C3** = [Cu(**6**)(**9**)]$^+$ | NO$_2$ | 0.778 | 2.93 | −16.7 |
| **C4** = [Cu(**6**)(**10**)]$^+$ | H | 0.000 | 4.10 | −23.4 |
| **C6** = [Cu(**7**)(**8**)]$^+$ | OMe | −0.268 | 4.63 | −26.4 |

[a] Substituent at 4-position of the pyridine head groups.

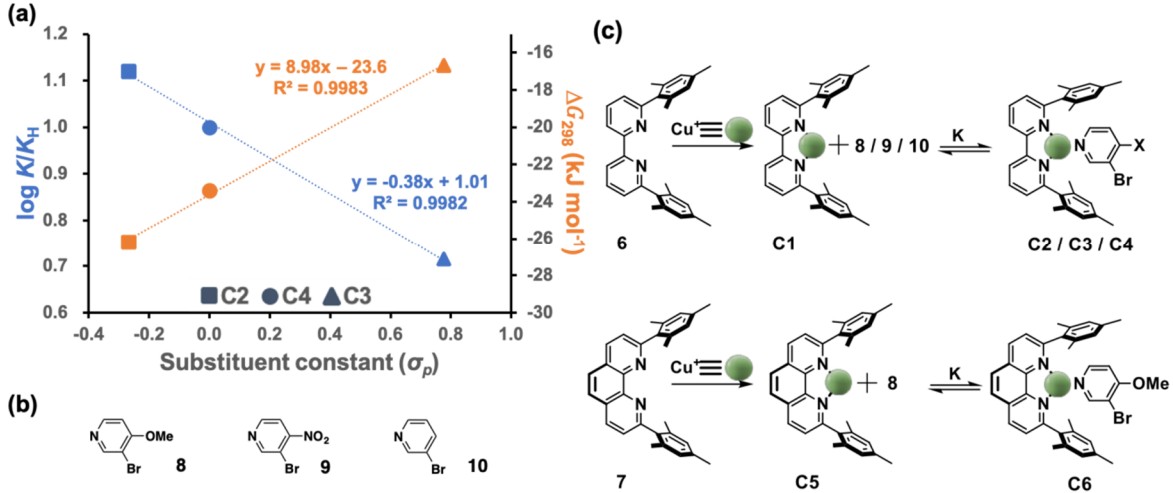

**Figure 2.** (**a**) The normalized binding constant log $K/K_H$ and the Gibbs free energy correlate linearly with the substituent constant. All binding constants were measured in CH$_2$Cl$_2$. (**b**) Chemical structures of substituted pyridines **8**, **9** and **10**. (**c**) Chemical structures of model complexes **C1–C6**.

### 3.2. Synthesis and Characterization of Four-Component Nanorotors

The strong correlation between $\sigma_p$ and log $K$ encouraged us to prepare the four-component nanorotors based on the above design of the head group. In this regard, the substituted pyridine rotators **1–3** were synthesized by Sonogashira coupling of zinc(II)-5-(4-ethynylphenyl)-10,15,20-trimesitylporphyrin (**11**) with **8**, **9** and **10**, as shown in Scheme 1.

The new bipyridine stator **4** was afforded in six steps, as outlined in Scheme 2. First, the bipyridine binding site was separately prepared in two steps: Suzuki–Miyaura coupling between 6,6′-dibromo-2,2′-bipyridine (**12**) and mesitylene-2-boronic acid (**13**)–furnished 6-bromo-6′-mesityl-2,2′-bipyridine (**16**) in 75% yield. On the other hand, nucleophilic substitution at isopropoxyboronic acid pinacol (**15**), using the lithiated species obtained from 1-iodo-2,3,5,6-tetramethyl-4-[2-(trimethylsilyl)ethynyl]-benzene (**14**), furnished compound **17**. Thereafter, Suzuki–Miyaura coupling between **16** and **17** yielded product **18**. A follow-up deprotection of the TMS-alkyne unit in the presence of NaCO$_3$ provided **19** in good yield. After Sonogashira coupling between **19** and zinc(II)-*meso*-5,15-bismesityl-10,20-bis(4-iodophenyl)porphyrin (**20**), the stator **4** was afforded in 45% yield (over five steps). The new ligands **1–4** were characterized unambiguously by NMR, ESI-MS, IR and elemental analysis (see Supplementary Materials).

**Scheme 1.** Synthesis of rotators **1–3**. (a) Pd(PPh₃)₄, DMF, Et₃NH, 80 °C, 14 h, 83%; (b) Pd(PPh₃)₄, THF, *i*-Pr₂NH, 70 °C, 14 h, 88%; (c) Pd(PPh₃)₄, THF, *i*-Pr₂NH, 70 °C, 14 h, 81%.

**Scheme 2.** Synthesis of stator **4**. (a) Pd(PPh₃)₄, H₂O-MeOH-THF, 90 °C, 12 h, 75%; (b) *n*-BuLi, THF, *n*-hexane, −78 °C, 1 h, then **15**, rt, 1 h, 80%; (c) Pd(PPh₃)₄, H₂O-THF, 90 °C, 12 h, 85%; (d) Na₂CO₃, MeOH/THF, rt, 14 h, 95%; (e) Pd(PPh₃)₄, THF, *i*-Pr₂NH, 75 °C, 14 h, 80%.

To prepare the prerotor **ROT-1a** = [Cu(**1**)(**4**)(DABCO)]⁺ (Figure 3), rotator **1** was mixed with stator **4**, DABCO and [Cu(CH₃CN)₄](PF₆) (1:1:1:1) in CD₂Cl₂. The ¹H NMR displayed two singlets (1:1 ratio) for protons of the bipyridine–mesityl unit, e.g., for 9-H at 7.01 and 6.92 ppm and 16-H at 2.38 and 2.33 ppm (Figure 3a). All bipyridine protons (3-H, 4-H, 5-H, 6-H, 7-H and 8-H) showed up as two sets of signals (1:1 ratio) as well. For protons [3/8]-H, the downfield-shifted set of signals corresponded to that of the coordinated bipyridine station, i.e., $N_{\mathrm{py}} \rightarrow$ [Cu(bipyAr₂)]⁺, while the upfield signals originated from the unloaded bipyAr₂ (Figure S15). Two sets of broad singlets at −4.45 and −4.54 ppm, corresponding to the DABCO protons, confirmed that DABCO acted as a hinge between the two different zinc porphyrins, **1** and **4**. The ¹H NMR shifts, thus, clearly confirmed formation of **ROT-1a** as a hetero-sandwich structure. ¹H-¹H COSY, ESI-MS, and elemental analysis further supported the assignment.

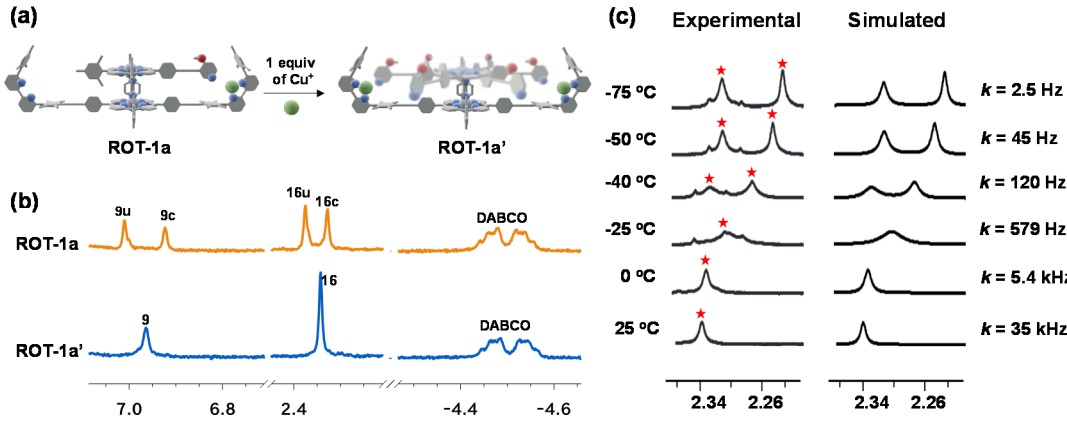

**Figure 3.** (**a**) Cartoon representation of **ROT-1a** and **ROT-1a′**. (**b**) Comparison of partial $^1$H NMR (CD$_2$Cl$_2$, 400 MHz, 298 K) of **ROT-1a** and **ROT-1a′**. Protons in **ROT-1a** that correspond to the heteroleptic pyridine and phenanthroline (HETPYP)-complexed bipyridine (9c-H, 16c-H) are different from those that belong to the unloaded bipyridine (9u-H, 16u-H) and the averaged signals (9-H, 16-H) in **ROT-1a′**. (**c**) Partial VT $^1$H NMR (CD$_2$Cl$_2$, 600 MHz) of **ROT-1a′** shows the splitting of the 16-H signal at low temperature (red asterisk).

After the addition of one further equivalent of [Cu(CH$_3$CN)$_4$](PF$_6$) to **ROT-1a**, the resulting **ROT-1a′** = [Cu$_2$(**1**)(**4**)(DABCO)]$^{2+}$ exhibited averaged signals for protons 9-H and 16-H (Figure 3b), as well as for all other bipyridine protons (Figure S17). Eventually the $^1$H-DOSY corroborated **ROT-1a′** as a single species in solution (Figure S39). The dynamic behavior in the nanorotor was monitored by variable temperature (VT) $^1$H NMR, in the range from 25 to −75 °C in CD$_2$Cl$_2$ (Figure 3c). During cooling, the signal of proton 16-H transformed from a sharp singlet at 25 °C (2.34 ppm) to a broad singlet at −25 °C (2.31 ppm) and then split into two singlets at −40 °C (2.32 and 2.27 ppm). These two singlets were assigned to the copper(I)-loaded bipyridine stations that were either connected to the pyridine terminal or not. The rotational frequency of the rotor was determined by simulation of the proton 16-H signal, using the WinD-NMR (Department of Chemistry, University of Wisconsin, Madison, WI, USA) software to $k_{298}$ = 3.5 × 10$^4$ Hz at 25 °C and the corresponding free energy activation barrier to $\Delta G^\ddagger_{298}$ = 47.1 kJ mol$^{-1}$ (Figure S32).

Preparation of the prerotors **ROT-1b** = [Cu(**1**)(**5**)(DABCO)]$^+$, **ROT-2** = [Cu(**2**)(**4**)(DABCO)]$^+$ and **ROT-3** = [Cu(**3**)(**4**)(DABCO)]$^+$ were accomplished in a similar manner as that of **ROT-1a**. Addition of one equiv of [Cu(CH$_3$CN)$_4$]PF$_6$ to **ROT-1b**, **ROT-2** or **ROT-3** afforded rotors **ROT-1b′** = [Cu$_2$(**1**)(**5**)(DABCO)]$^{2+}$, **ROT-2′** = [Cu$_2$(**2**)(**4**)(DABCO)]$^{2+}$ and **ROT-3′** = [Cu$_2$(**3**)(**4**)(DABCO)]$^{2+}$, respectively. Analysis of the VT $^1$H NMR (Figures S31–S38) revealed an exchange frequency at 298 K of $k_{298}$ = 2.0 × 10$^4$ Hz and activation free energy barrier $\Delta G^\ddagger_{298}$ = 48.6 kJ mol$^{-1}$ for **ROT-1b′**; $k_{298}$ = 8.4 × 10$^5$ Hz, $\Delta G^\ddagger_{298}$ = 39.2 kJ mol$^{-1}$ for **ROT-2′**; and $k_{298}$ = 7.7 × 10$^4$ Hz, $\Delta G^\ddagger_{298}$ = 45.2 kJ mol$^{-1}$ for **ROT-3′**. The rotational frequencies of the nanorotors showed significant differences (Table 2).

**Table 2.** Experimental rotational frequency and activation barriers at 25 °C, as calculated from VT $^1$H NMR.

| Nanorotor | 4-X | $\Delta G^\ddagger_{298}$ (kJ mol$^{-1}$) | $k_{298}$ (Hz) | log $k$ |
|---|---|---|---|---|
| **ROT-1a′** | 4-OMe | 47.1 | 3.5 × 10$^4$ | 4.55 |
| **ROT-1b′** | 4-OMe | 48.6 | 2.0 × 10$^4$ | 4.30 |
| **ROT-2′** | 4-NO$_2$ | 39.2 | 8.4 × 10$^5$ | 5.93 |
| **ROT-3′** | 4-H | 45.2 | 7.7 × 10$^4$ | 4.88 |

## 4. Discussion

### 4.1. Bipyridine vs. Phenanthroline Stator in Nanorotors

By comparing **ROT-1a′** with **ROT-1b′**, the rotational frequency difference between bipyridine stator and phenanthroline stator turned out to be rather small ($k_{298}$ of **ROT-1a′**

and **ROT-1b′** = $3.5 \times 10^4$ Hz and $2.0 \times 10^4$ Hz, respectively). The coalescence temperature of **ROT-1a′** and **ROT-1b′** in the VT $^1$H-NMR was also located in the same range, around $-25\,°C$. The similar binding constant of **C2** and **C6** is in full agreement with the kinetic finding.

The minor rate difference between the two rotors suggests that it is mainly the pyridine → copper(I) interaction that matters in the rate-determining step. As one would expect on the basis of the higher flexibility of the bipyridine ligand, **ROT-1a′** (bipyridine station) rotates a little bit faster than **ROT-1b′**. We assume that the bipyridine ligand is more apt to adjust to the distortions at the chelate binding site in the transition state [42,43].

### 4.2. Hammett Equation Applies to Rotational Exchange in Nanorotors

The plot of the nanorotors' rotational frequency log $k/k_H$ and of $\Delta G^{\ddagger}_{298}$ of **ROT-1a′**, **ROT-2′** and **ROT-3′** against $\sigma_p$ revealed a linear correlation (Figure 4a), indicating that the electronic effect of the *para*-substituent is the major contributing factor in the rate-determining step and that other effects are negligible. Due to the aforementioned linear relationship between log $K/K_H$ of complex formation for **C1**, **C2** and **C3** and $\sigma_p$, a linear free energy relationship (LFER) between thermodynamic and kinetic data was established (Figure 4b), suggesting that the Hammett equation can directly be used for predicting the rotational frequency in nanorotors with other *para*-substituents.

$$\log k/k_H = 0.27\,\sigma_p + 1.00\ (R^2 = 0.9997) \tag{3}$$

$$\Delta G^{\ddagger}_{298}\ (kJ\ mol^{-1}) = -7.59\,\sigma_p + 45.12\ (R^2 = 0.9997) \tag{4}$$

$$\log k_{298} = -0.84\,\log K + 8.39\ (R^2 = 0.9964) \tag{5}$$

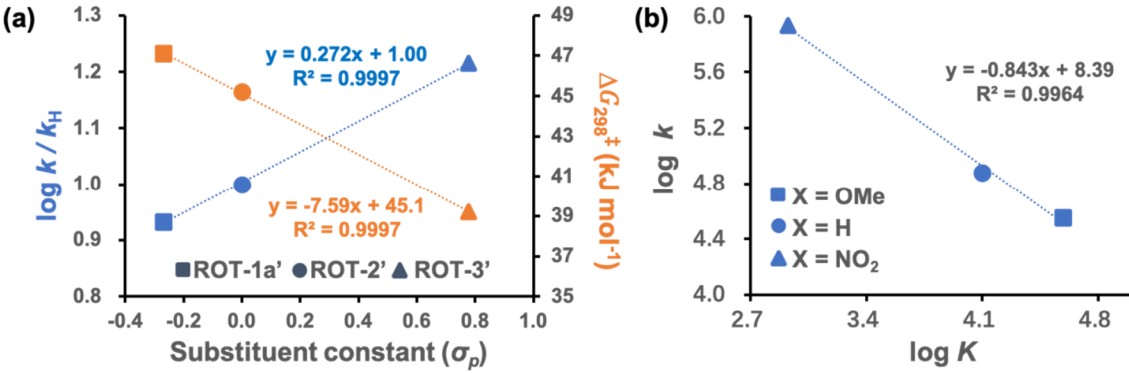

**Figure 4.** (**a**) The rotational frequency log $k/k_H$ and the activation Gibbs free energy correlate linearly with the substituent constant. (**b**) The correlation between binding constants of **C2**, **C3** and **C4** and the rotational frequencies log $k$ of **ROT-1a′**, **ROT-2′** and **ROT-3′**.

One therefore expects that substituents like 4-NMe$_2$ ($\sigma_p = -0.600$) and 4-NMe$_3^+$ ($\sigma_p = 0.860$) would result in extreme cases of the rotational frequency. By using the correlation log $k_{298} = 1.33\,\sigma_p + 4.89$ (Figure S62) for the rotor with the rotator head 4-NMe$_2$, we would expect log $k_{298} = 4.09$, $k_{298} = 1.2 \times 10^4\ s^{-1}$ and $\Delta G^{\ddagger}_{298} = 51.7\ kJ\ mol^{-1}$, whereas, for that with rotator head 4-NMe$_3^+$, the expected kinetic data are log $k_{298} = 6.03$, $k_{298} = 1.1 \times 10^6\ s^{-1}$ and $\Delta G^{\ddagger}_{298} = 38.6\ kJ\ mol^{-1}$.

## 5. Conclusions

In conclusion, we herein presented a small series of four-component nanorotors for evaluating conformational and electronic effects on their rotational exchange frequency. Whereas the effect of higher flexibility at the bipyridine vs. phenanthroline binding site in the stator was rather small (less than a factor of 2), the variation of the electronic character of substituents in the *para*-position of the pyridine head group in the rotator led to distinct

differences in rotational speed (almost 25-fold). The rotational speed, as well as the Gibbs free activation energy, shows excellent linear correlation with the Hammett substituent constant. Based on the Hammett equation, the rotational frequency of any analogously substituted nanorotor can be predicted, resulting in a 100-fold change of the exchange frequency (from -NMe$_2$ to -NMe$_3{}^+$). Since the exchange frequency of this type of four-component rotor is correlated with the rate of click catalysis, as established recently [37], one should be able to more extensively test the concept that rotating catalytic machinery shows reduced product inhibition at higher machine speed.

**Supplementary Materials:** The following are available online at https://www.mdpi.com/2624-854 9/3/1/9/s1. Supporting information containing all synthetic procedures, NMR spectra, VT-NMR and binding studies.

**Author Contributions:** Synthesis and characterization of ligands **1**, **2** and **4** and rotors by Y.-F.L.; VT simulation, binding constant measurement and ESI-MS measurements by A.G.; synthesis of ligand **3** by S.S.; synthesis of ligand **5** by P.K.B.; writing of the draft by Y.-F.L.; conception of the project and perfection of the manuscript by M.S. All authors have read and agreed to the published version of the manuscript.

**Funding:** This research was generously funded by the Deutsche Forschungsgemeinschaft (DFG), grant number Schm 647/20-2, and supported by the University of Siegen.

**Data Availability Statement:** The data presented in this study are available in the article and in the Supplementary material.

**Acknowledgments:** We gratefully acknowledge the support by Thomas Paululat (Siegen) in the measurement of the VT $^1$H NMR spectra.

**Conflicts of Interest:** The authors declare no conflict of interest.

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
