# Peer review of "Exchange Speed of Four-Component Nanorotors Correlates with Hammett Substituent Constants†"

_chemistry, doi:10.3390/chemistry3010009_

Round 1

Reviewer 1 Report

The manuscript by Schmittel and co-workers presents Hammett constant correlation with rotational frequency of nanorotors. The result is a valuable piece of science and will. e much appreciated by the community, however the manuscript requires a major revision prior acceptance.

Conceptual:

  1. Data looks very reliable, however, one can take three points and fit them with nearly any function; linear will be the only one hard to get if obviously point are not on the same line. This reviewer believes that one more post to the final data plot is extremely important to obtain. The synthetic protocol seems and claims to be very robust, thus I do not foresee much issues. E.g. 3-bromo-4-methylpyridine or 3-bromo-4-chloropyridine are readily available substrates and any of them can be utilized to obtain one more rotor to enhance this study. I believe lab has a stock of compound 11 and Sonogashira coupling will not make any problem with either recommended substrate. Made predictions about NMe2 and NMe3+ are very nice, but probably require a little of more experimental support. The authors, e.g. for VT-NMR present more than 3 points to obtain a dependency.

Major:

  1. By definition Hammett equation represents relative dependence, thus I highly recommend to plot diagrams as log(k/k(H)), not simply log(k). I agree and understand that result will remain the same, however, this representation is much clearer description of the observed correlation.
  2. Phase correction for most of 1H spectra is extremely poor. All spectra has to be fixed prior publication. E.g. Figures S1, S2, S3, S19!

Minor:

  1. In the SI, for the mass spectra, captions are moved to the other pages than figures.
  2. Blue line of theoretical isotope pattern should be explained in the figure captions.
  3. I do not want to downplay the authors contributions, but one would like to recommend adding another 10 papers to the reference list to make self citation of ~20% rather than 25%. Unnecessary citation is not good either, thus I do not see an easy solution and all cited research is highly  important and relevant to this study.

Reviewer 2 Report

The paper described the supramolecular nanorotors with various pyridine head groups and copper(I) stopper. It is remarkable that the difference of the para-substituents at the pyridine head group of the rotator led to distinct differences in activation free energy with an excellent linear correlation with Hammett substituent constants. The experiments for synthesis, characterization, and dynamics analysis were carried out very well. I think the paper could be published in the Chemistry after minor corrections. My opinions are described as follows:

  1. In page 6 and table 2, the author compared the activation free energies for the rotation of nanorotors. According to the Eyring plots shown in SI, these free energy difference was arising from the difference of the activation entropies rather than activation enthalpies. I think this is surprising for readers. If you have any comments for the phenomena, please describe in the main text. Additionally, please correct the symbol  “#” shown in the plots in the SI.
  2. In the abstract, the authors addressed that the strength of the copper(I)-ligand interaction is the key determining factor in the rate-determining step. I wonder this sentence causes misunderstandings, because the activation enthalpies for the rotation showed no substituent dependence.
  3. In page 6, line 163, the author discussed the NMR signals of 9-H and 16-H. Although the atom numbering was shown in SI pictures, there is no such figure in the main text. Please specify the positions using structural fomula clearly.
  4. In page 5, scheme 2, there are several difference in the synthetic conditions between the picture and the caption, for instance, the condition (f) was only shown in the caption. Please check and correct.
  5. In page 4, Figure 2, please indicate name of the solvent for the experiments in the figure or the caption.
  6. In page 8, line 228, the authors described “a 100-fold change of the exchange frequency (from -NMe2 to -NMe3+)”. I think the parentheses are unnecessary.

Reviewer 3 Report

This paper reports the relation between the Hammett substituent constants and the rate of the rotation of assembled rotors with a different substituent on the pyridyl group coordinating to the Cu(I) center and shows a linear free energy relationship between thermodynamic with kinetic data in molecular motion. All the molecules were well characterized by several spectroscopies and other way(s) and the kinetic parameters were determined by the line-shape analysis of 1H NMR spectra obtained by variable temperature 1H NMR spectroscopy. The liner relation between the Hammett parameters and the formation constant (and the free energies) of the complexation was confirmed for the model compound. Then the linear free energy relationship was investigated by the molecular rotors with different substituents. The results and the conclusion are simple but is useful for further molecular design, which is valuable for the researchers in this community. Thus, I recommend the publication of this paper in this journal as it is.

Round 2

Reviewer 1 Report

Work was improved and can be now accepted